# Ribosome biogenesis restricts innate immune responses to virus infection and DNA

Christopher Bianco[1], Ian Mohr[1,2]*

[1]Department of Microbiology, NYU School of Medicine, New York, United States; [2]Laura and Isaac Perlmutter Cancer Institute, NYU School of Medicine, New York, United States

**Abstract** Ribosomes are universally important in biology and their production is dysregulated by developmental disorders, cancer, and virus infection. Although presumed required for protein synthesis, how ribosome biogenesis impacts virus reproduction and cell-intrinsic immune responses remains untested. Surprisingly, we find that restricting ribosome biogenesis stimulated human cytomegalovirus (HCMV) replication without suppressing translation. Interfering with ribosomal RNA (rRNA) accumulation triggered nucleolar stress and repressed expression of 1392 genes, including High Mobility Group Box 2 (HMGB2), a chromatin-associated protein that facilitates cytoplasmic double-stranded (ds) DNA-sensing by cGAS. Furthermore, it reduced cytoplasmic HMGB2 abundance and impaired induction of interferon beta (IFNB1) mRNA, which encodes a critical anti-proliferative, proinflammatory cytokine, in response to HCMV or dsDNA in uninfected cells. This establishes that rRNA accumulation regulates innate immune responses to dsDNA by controlling HMGB2 abundance. Moreover, it reveals that rRNA accumulation and/or nucleolar activity unexpectedly regulate dsDNA-sensing to restrict virus reproduction and regulate inflammation. (*145 words*)

**\*For correspondence:**
ian.mohr@med.nyu.edu

**Competing interests:** The authors declare that no competing interests exist.

## Introduction

The minting of ribosomes, the molecular machines that synthesize polypeptides, is subject to stringent controls that coordinate demand for ribosomes with intracellular processes and environmental cues, including nutritional status and responses to physiological stress (*Iadevaia et al., 2014*). It is among the most energetically demanding processes in the cell and requires all three eukaryotic RNA polymerases. While the 47S precursor-rRNA (pre-rRNA) is transcribed by RNA Polymerase I (RNAPI) within the nucleolus, 5S rRNA is generated by RNA Polymerase III (RNAPIII) and mRNAs encoding ribosomal proteins are produced by RNA Polymerase II (RNAPII) in the nucleoplasm. Following processing of 47S pre-rRNA into mature 5.8S, 18S, and 28S rRNA, assembly of large 60S and small 40S ribosomal subunits from rRNA and ribosomal protein components commences in the nucleolus (*Mayer and Grummt, 2006*). While ribosome biogenesis is fundamental for normal cell growth, development and differentiation, its dysregulation contributes to diverse diseases including cancer, where it is stimulated, and genetic disorders collectively termed ribosomopathies where it is often impaired (*Aspesi and Ellis, 2019*; *Mills and Green, 2017*; *Narla and Ebert, 2010*; *Pelletier et al., 2018*). Interfering with ribosome biogenesis triggers a nucleolar stress response whereby p53 is stabilized and the cell cycle is blocked (*Boulon et al., 2010*; *Chakraborty et al., 2011*; *Deisenroth et al., 2016*; *Golomb et al., 2014*). Furthermore, ribosomes play critical roles in infection biology where polypeptide synthesis by obligate intracellular parasites such as viruses is absolutely reliant upon cellular ribosomes (*Mohr and Sonenberg, 2012*; *Stern-Ginossar et al., 2019*). Indeed, several viruses reportedly stimulated rRNA synthesis upon infection (*May et al.,*

*1976*; *Pöckl and Wintersberger, 1980*; *Soprano et al., 1983*). However, whether the ribosome itself or the process of ribosome biogenesis might be subverted by viruses or incorporated into the cellular repertoire of innate immune responses remains largely unexplored. Moreover, while virus-induced ribosome biogenesis is presumed to benefit infection by ensuring ribosome sufficiency for protein synthesis, this hypothesis remains untested. In this investigation we not only challenge this widely held conviction, but reveal how ribosome production is unexpectedly integrated into innate responses that regulate inflammatory cytokine induction.

Infection with Human cytomegalovirus (HCMV) provides a powerful model system to evaluate the impact of ribosome biogenesis on inflammatory responses and virus infection biology. HCMV is a large DNA virus that replicates in the nucleus and although wide-spread and commonly asymptomatic, infection results in life-threatening disease among the immunocompromised and remains a major source of congenital morbidity and mortality among newborns (*Boeckh and Geballe, 2011*; *Britt, 2008*; *Cannon et al., 2010*; *Ljungman et al., 2010*; *Manicklal et al., 2013*; *Razonable et al., 2013*). In contrast to viruses that shut off host mRNA translation, host protein synthesis proceeds in cells infected with Human Cytomegalovirus (HCMV) (*Walsh et al., 2005*). This potentially permits protein products encoded by antiviral mRNAs to accumulate (*Bianco and Mohr, 2017*) while viral and host mRNAs must compete for a limited supply of ribosomes. To mitigate cellular antiviral immune responses without globally impairing translation of host mRNAs, HCMV relies upon multiple independent mechanisms (*Biolatti et al., 2018*; *Li et al., 2013*; *Paulus and Nevels, 2009*). One of these involves a virus-directed increase in the abundance of host translation factors, suggesting a viral mechanism to facilitate simultaneous synthesis of host and viral proteins (*Tirosh et al., 2015*; *McKinney et al., 2014*; *Perez et al., 2011*; *Walsh et al., 2005*). Indeed, HCMV-induced increases in host translational machinery have been shown to be required for efficient viral replication (*McKinney et al., 2012*). Moreover, HCMV upregulates the translation of mRNAs encoding functions involved in ribosome biogenesis and increases the steady state levels of ribosomes (*McKinney et al., 2014*; *Tirosh et al., 2015*). Although untested, it was believed that this increase in ribosome abundance is enforced by the virus and required to foster efficient translation of both host and viral mRNAs to facilitate viral replication.

Here, we test this hypothesis by blocking ribosome biogenesis in HCMV-infected cells. Surprisingly, while limiting ribosome biogenesis impairs global protein synthesis in uninfected cells, it has no detectable impact in HCMV-infected cells, suggesting that ribosomes are not rate limiting components and that their increase in HCMV infected cells is not required for efficient protein synthesis. Furthermore, not only was HCMV productive growth insensitive to restricting ribosome biogenesis, it was unexpectedly substantially enhanced. In ribosome biogenesis impaired cells, RNA-seq identified a selective decrease in the abundances of genes whose transcription is augmented by Nuclear Factor-Y (NF-Y) and repressed by the DREAM complex. One of these genes, High Mobility Group Box 2 (HMGB2), is repressed by p53 and its product reportedly plays a role enhancing cell intrinsic immune responses by augmenting dsDNA detection to produce an antiviral cellular state (*Yanai et al., 2009*; *Yanai et al., 2012*). When ribosome biogenesis is impaired, IFNB1 mRNA accumulation in response to HCMV is compromised and ISG mRNAs and proteins accumulate to lower levels. Furthermore, while introduction of synthetic dsDNA, the major pathogen associated molecular pattern (PAMP) delivered by HCMV, into cells induces a robust innate immune response, this response is hampered when ribosome biogenesis is inhibited, suggesting a central role for ongoing ribosome production in the cellular immune response to a viral PAMP. This, for the first time, establishes a connection between nucleolar activity, ribosome abundance, and cell intrinsic immunity.

## Results

### RNAPI transcription factor abundances and RNAPI activity are stimulated by HCMV

Due to the ability of RNAPI to coordinate accumulation of all ribosome components and drive ribosome biogenesis (*Laferté et al., 2006*), its properties in HCMV-infected cells were investigated. Earlier studies (*Tanaka et al., 1975*) indicated that 45S pre-rRNA synthesis was stimulated in response to HCMV infection at 30 hr post-infection (hpi). To substantiate this finding and investigate the kinetics of rRNA accumulation, the overall abundance of cellular 45S rRNA precursor was monitored in

response to HCMV infection over time. *Figure 1A* shows that increased 45S rRNA levels were detected in HCMV-infected compared to mock-infected cells as early as three hpi. This continued to increase, reaching its zenith at 24 hpi where 45S rRNA levels were approximately 6-fold greater in HCMV-infected cells than mock-infected cells. By 48 hr, 45S rRNA abundance declined slightly but remained elevated relative to levels in mock-infected cells through 72 hpi (*Figure 1A*). Consistent with increased 45S rRNA accumulation, immunostaining for fibrillarin, a component marking fibrillar centers of nucleoli (*Németh and Grummt, 2018*) revealed that nucleolar size increased in HCMV-infected cells and that nucleolar integrity appeared intact (*Figure 1B*). Since rRNA is highly transcribed, analysis of newly synthesized RNA by 5-Fluoruridine (FU) labeling followed by

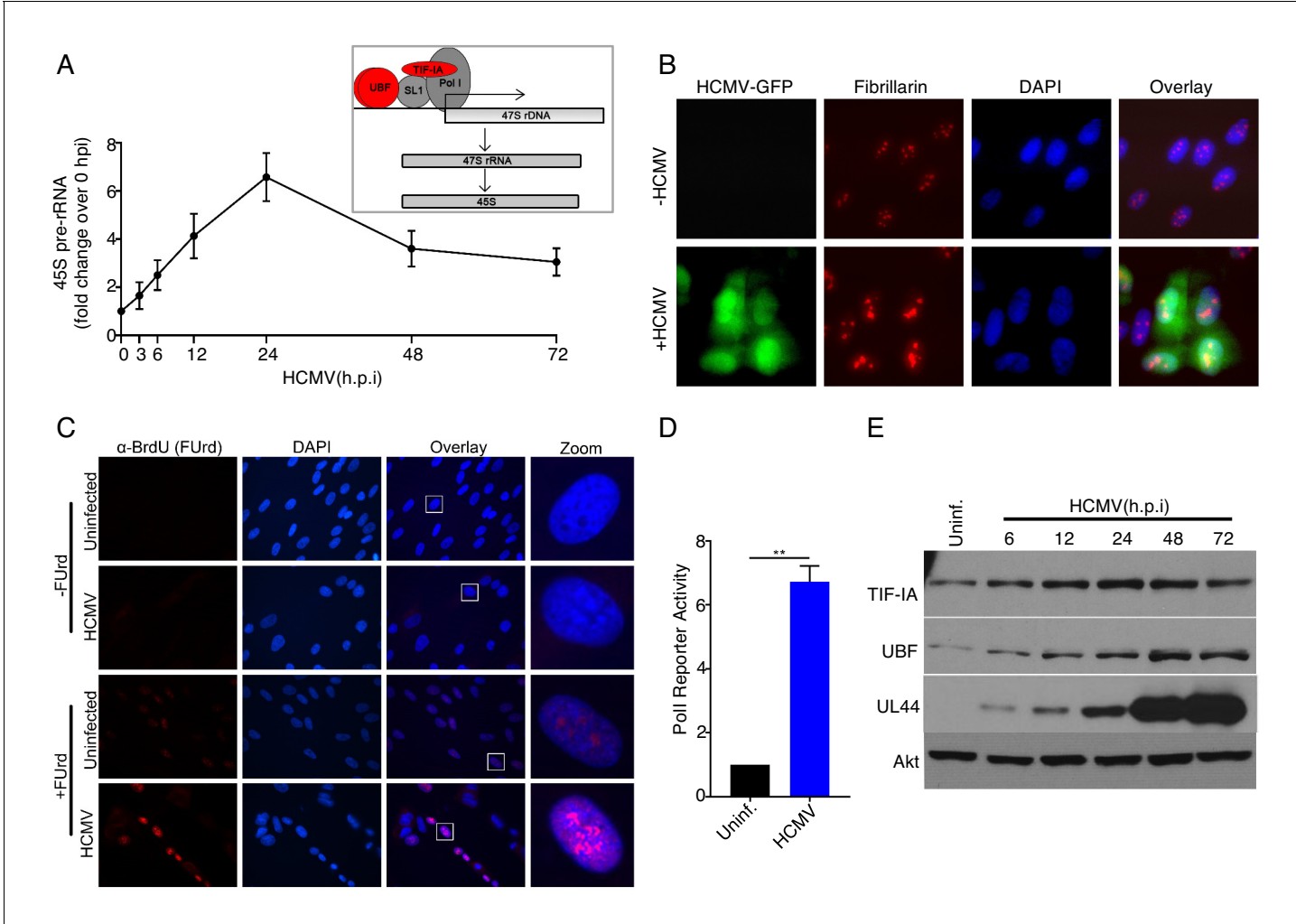

**Figure 1.** Regulation of RNA polymerase I activity by HCMV in human fibroblasts. (a) Growth arrested NHDFs were mock- or HCMV-infected (MOI = 3 PFU/cell). Total RNA was isolated at the indicated time points, and RT-qPCR was performed using primers specific for 45S pre-rRNA (n = 4) *Inset box*: Production of 45S pre-rRNA. Cellular factors involved in RNA polymerase I (Pol I) transcription of DNA that encodes the full length 47S rRNA precursor are depicted (47S rDNA; horizontal arrow indicates direction of transcription). Pol I specific transcription factors (TIF-IA, UBF) referred to throughout the manuscript appear in red. The resulting 47S full length rRNA precursor product which is processed into 45S pre-rRNA are shown below. (b) NHDFs were mock-infected or HCMV infected (MOI = 3 PFU/cell) and fixed in 4% PFA 48hpi. Immunofluorescence staining was performed using an antibody specific for fibrillarin. Signal in the FITC channel represents the HCMV produced eGFP reporter (n = 2). (c) 48hpi, mock- or HCMV- infected NHDFs were labeled with 2 mM 5'-Fluorouridine for 20 min. prior to fixation in 4% PFA. Immunofluorescence staining was performed using an antibody specific for BrdU (n = 2). Rectangle in overlay panel indicates nucleus shown in zoom panel. (d) Growth arrested NHDFs were transfected with pHrP2-BH reporter plasmid. After 24 hr, cells were mock- or HCMV-infected. 24hpi, total RNA was isolated, and RT-qPCR was performed using primers specific for the pHrP2-BH reporter transcript. The error bars indicate SEM. **p≤0.01; Student's *t* test. (e) Total protein from NHDFs mock infected or infected with HCMV (MOI = 3 PFU/cell) was collected at the indicated times, fractionated by SDS-PAGE, and analyzed by immunoblotting using antibodies specific for TIF-IA, UBF, UL44, and Akt (loading control) (n = 2).

immunofluorescence to detect metabolically labeled RNA can be used to examine nascent rRNA accumulation and subcellular distribution. *Figure 1C* demonstrates that FU incorporation was stimulated and areas of FU staining were enlarged in response to HCMV infection. Under these conditions, the majority of FU staining was confined within discrete sub-nuclear foci in uninfected or HCMV-infected cells (*Figure 1C*). These sub-nuclear sites of FU incorporation resembled nucleoli, which suggested that HCMV infection might stimulate the cellular RNA polymerase I (RNAPI). To determine if the increase in 45S pre-rRNA reflected increased RNAPI activity or a decrease in 45S pre-rRNA processing into mature rRNA, transcript accumulation from a reporter plasmid containing an RNAPI-specific promoter was measured in mock-infected and HCMV-infected cells (*Mayer et al., 2005*). In response to HCMV infection, a significant increase in RNAPI reporter activity was detected at 24 hpi (*Figure 1D*), demonstrating that HCMV stimulates transcription from cellular RNAPI-specific promoters. It further indicates that HCMV infection stimulates RNAPI.

To define how HCMV infection might stimulate RNAPI transcription, total protein isolated from mock-infected or HCMV-infected cells was analyzed by immunoblotting and overall levels of the RNAPI specific transcription factors TIF-IA and UBF monitored. Compared to mock-infected cells, the abundance of the RNAPI transcription factors TIF-IA and UBF increased by six hpi coincident with detection of UL44, a representative early viral protein (*Figure 1E*). TIF-IA and UBF reached peak levels by 48hpi, and remained elevated even at 72hpi (*Figure 1E*). This raised the possibility that HCMV infection might drive RNAPI transcriptional activity by increasing RNAPI transcription factors abundance.

## Ribosome abundance and protein synthesis can be uncoupled in HCMV-infected cells

To determine if the virus-induced increase in TIF-IA abundance was required to stimulate 45S pre-rRNA accumulation, the impact of TIF-IA depletion on 45S pre-rRNA steady state levels and ribosome biogenesis in HCMV-infected cells was investigated. Following transfection of NHDFs with control non-silencing siRNA or one of two different siRNAs targeting TIF-IA, cells were infected with HCMV. Compared to non-silencing siRNA, both TIF-IA siRNAs effectively reduced 45S pre-rRNA steady-state levels in HCMV infected cells (*Figure 2A*). Sucrose gradient fractionation of cytoplasmic lysates from HCMV-infected cells revealed a substantial decrease in 40S and 60S ribosomal subunits and 80S monoribosomes in cells treated with TIF-IA specific siRNA compared to control siRNA (*Figure 2B*). However, a more modest impact on actively translating polyribosomes was observed, indicating that polysome assembly proceeds in HCMV-infected cells even when ribosome biogenesis is restricted (*Figure 2B*). It further suggested that the large reduction in ribosome subunits and monosomes in response to TIF-IA depletion may not impact ongoing protein synthesis in HCMV-infected cells. To address this possibility, control non-silencing and TIF-IA siRNA-treated cells were mock- or HCMV-infected and incubated with $^{35}$S-containing amino acids to label newly synthesized proteins (*Figure 2C,D*). Unexpectedly, while TIF-IA depletion impaired global protein synthesis in uninfected cells (*Figure 2C,E*), it had no detectable impact in HCMV-infected cells (*Figure 2D,F*). This indicated that the HCMV-induced increase in ribosome abundance, which was presumed important for the infected cell protein synthesis program, is dispensable for efficient protein synthesis during infection and that ribosome abundance per se was not limiting infected cell protein synthesis. While the precise molecular mechanism underlying this finding remains unknown, it clearly reveals unexpected differences in how ongoing protein synthesis responds to interfering with ribosome biogenesis in uninfected compared to HCMV-infected primary human fibroblasts.

## Interfering with ribosome biogenesis at multiple steps enhances HCMV replication

Having found that ribosome biogenesis, but surprisingly not overall protein synthesis was dependent upon the virus-induced increase in RNAPI-specific transcription factor TIF-IA abundance (*Figure 3A*), the impact of inhibiting ribosome biogenesis on HCMV acute reproduction was next investigated. Following transfection of control, non-silencing siRNA or TIF-IA targeting siRNAs, NHDFs were infected with HCMV expressing an eGFP reporter at a low multiplicity of infection (MOI). Compared with control siRNA-treated cultures, a greater number of eGFP-positive cells were detected in TIF-IA-depleted cultures (*Figure 3B*), indicating that TIF-IA depletion enhanced virus reproduction and

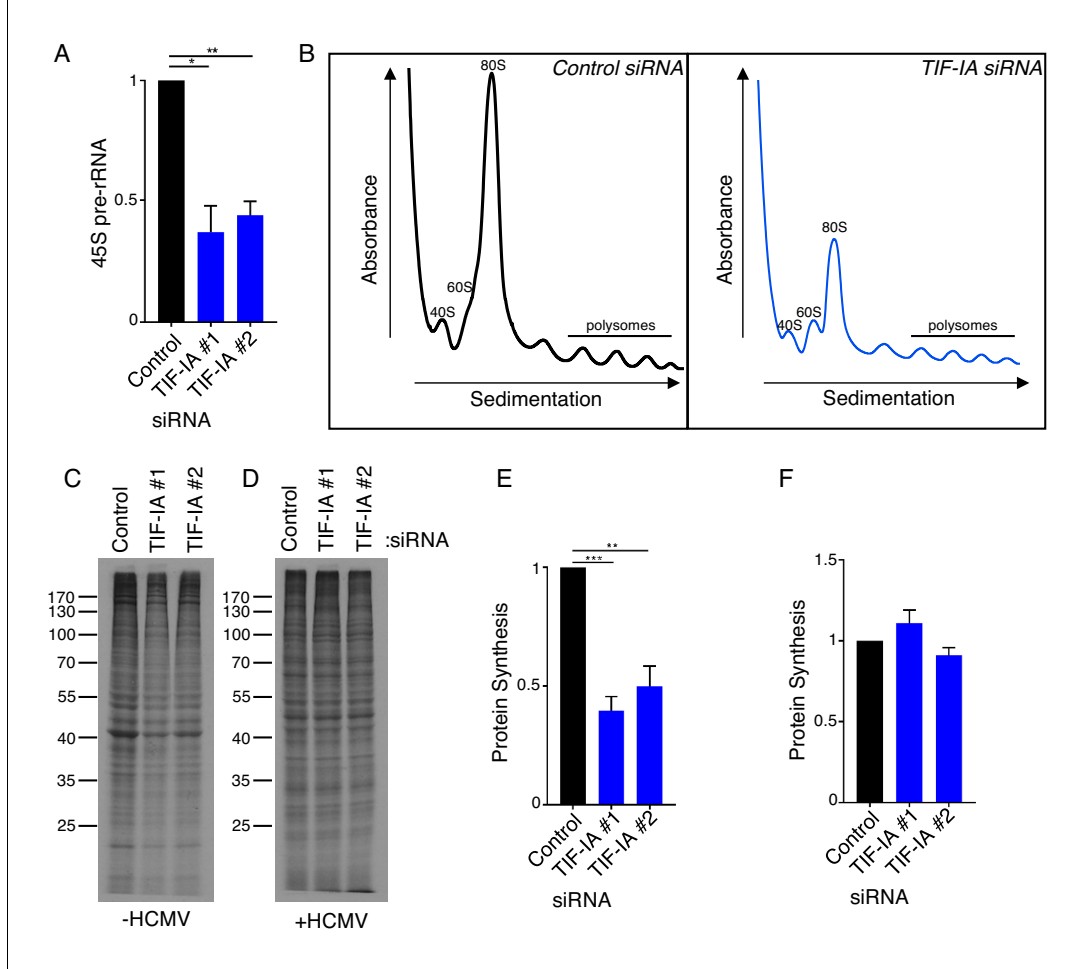

**Figure 2.** Ribosome biogenesis and protein synthesis are uncoupled in HCMV infected cells. (a) NHDFs were transfected with non-silencing (ns) control or TIF-IA siRNAs and infected with HCMV (MOI = 3 PFU/cell). At two dpi, total RNA was isolated, and RT-qPCR was performed using primers specific for 45S pre-rRNA. The error bars indicate SEM. *p≤0.05; **p≤0.01; Student's *t* test (n = 3). (b) As in (a), except cytoplasmic lysate was fractionated over a 10–50% sucrose gradient and the absorbance at 254 nm was recorded. The top of the gradient is on the left. (c–f) NHDFs were transfected with ns or TIF-IA siRNAs and (c,e) mock infected or (e,f) infected with HCMV (MOI = 3 pfu/cell) two dpi, cells were radiolabeled with $^{35}$S-amino acids and radioactive amino acid incorporation was (c,d) visualized by SDS-PAGE followed by autoradiography and (e,f) quantified by counting in liquid scintillant (n = 4). Error bars indicate SEM. **p≤0.01; ***p≤0.001; Student's *t* test.

dissemination. Immunoblotting of cellular lysates revealed that the abundance of representative early (UL44), and late (pp28) viral proteins all increased substantially upon TIF-IA depletion (*Figure 3C*). Furthermore, TIF-IA depletion enhanced infectious virus production by up to 100-fold (*Figure 3D*). These results establish that HCMV replication is restricted by the host RNAPI transcription factor TIF-IA.

To establish whether the enhancement of HCMV replication by TIF-IA depletion was selective for this specific RNAPI transcription factor, we repeated the experiment using siRNA specific for a different RNAPI transcription factor, UBF (*Figure 3A*). Similar to results obtained following TIF-IA depletion, greater numbers of eGFP-positive cells, more viral protein accumulation, and greater amounts of infectious virus were produced in HCMV-infected cultures upon UBF-depletion (*Figure 3E,F,G*).

While TIF-IA and UBF are required for 45S rRNA synthesis, they specifically target RNAPI. To determine if cellular factors critical for ribosome biogenesis other than those that specifically regulate RNAPI could influence HCMV replication, the impact of depleting the RNAPIII transcription factor TFIIIA, which is required for transcription of 5S rRNA (*Camier et al., 1995*; *Ciganda and Williams, 2011*), or BOP1, a rRNA processing factor required for maturation of 28S and 5.8S rRNA (*Strezoska et al., 2000*) was investigated (*Figure 3A*). Compared to control, non-silencing siRNA-

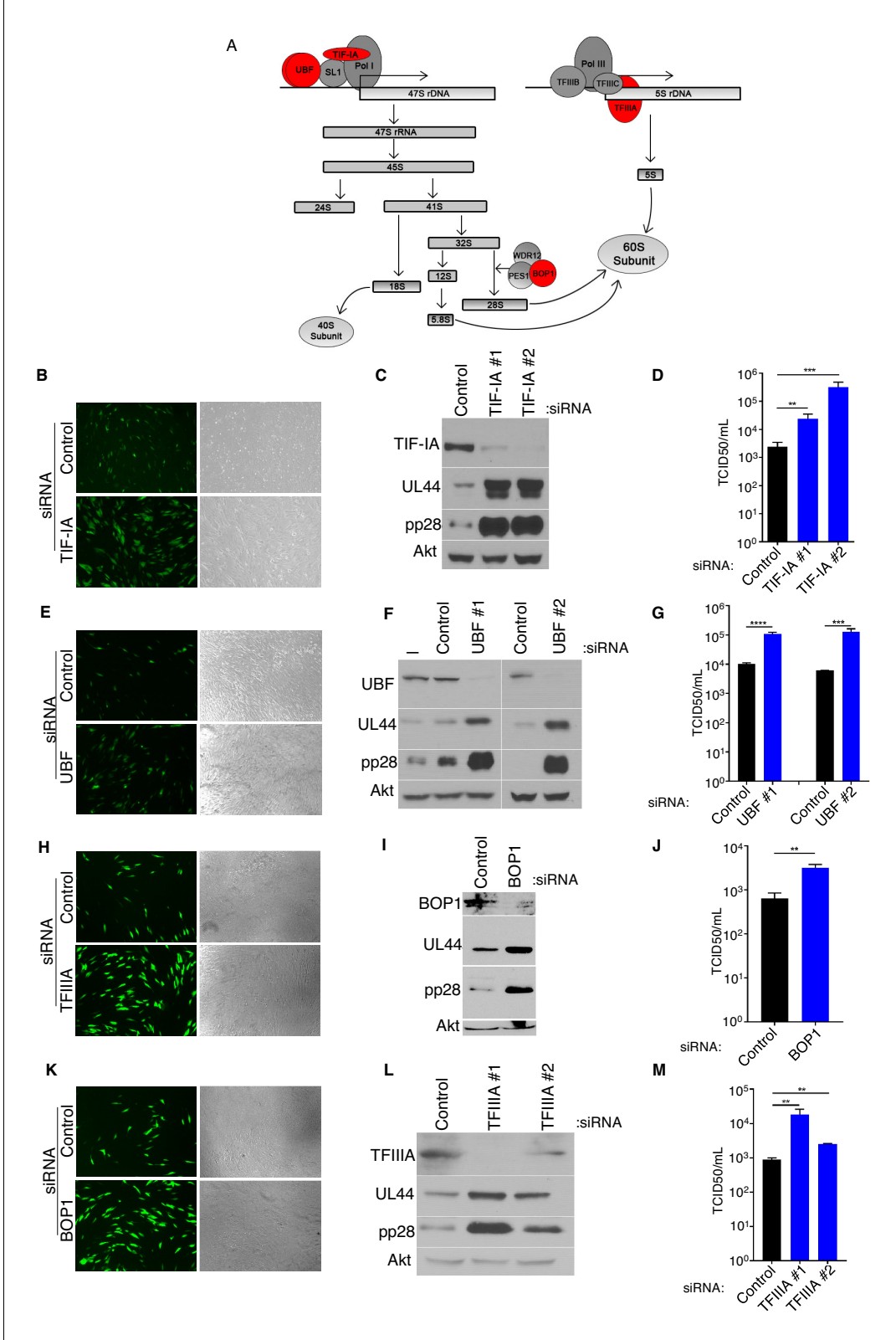

**Figure 3.** Inhibition of ribosome biogenesis enhances HCMV replication. (a) Diagram of rRNA biogenesis. Factors targeted in this study are shown in red. (b–k) NHDFs were transfected with non-silencing (ns) control, TIF-IA, UBF, TFIIIA, or BOP1 siRNAs, infected with HCMV (MOI = 0.05 pfu/cell) and (b,e,h,k) six dpi, photographs of eGFP fluorescence were captured to visualize virus replication, (c,f,i,l) total protein was collected, fractionated by SDS-PAGE, and analyzed by immunoblotting using antibodies specific for UL44, pp28, Akt (loading control), and (c) TIF-IA, (f) UBF, (i) BOP1, (l) TFIIIA. (d,g,j,

*Figure 3 continued on next page*

*Figure 3 continued*

m) Infectious virus was quantified from supernatants using a TCID50 assay. The error bars indicate SEM. *p≤0.05; **p≤0.01; ***p≤0.001; Student's *t* test. (n = 3).

The online version of this article includes the following figure supplement(s) for figure 3:

**Figure supplement 1.** Depletion of TFIIIA or BOP1 reduce ribosome abundances In HCMV infected cells.

treated cells infected with HCMV, levels of ribosomes in both TFIIIA and BOP1 siRNA-treated cells were reduced, consistent with these factors being required for HCMV-induced ribosome biogenesis (*Figure 3—figure supplement 1*). Furthermore, greater numbers of eGFP-positive cells, higher levels of representative viral proteins and increased infectious virus production were observed in cultures treated with TFIIIA or BOP1-siRNAs compared to non-silencing, control siRNA (*Figure 3H–M*). This demonstrates that interfering with 5S RNA transcription by depleting the RNAPIII specific transcription factor TIFIIIA or preventing processing of rRNA precursors by depleting BOP1 stimulates HCMV replication. It further shows that similar increases in HCMV productive growth are achieved by interfering with transcription factors selective for RNAPI or RNAPIII, or rRNA processing factors. Given that these very different factors target discrete processes essential for ribosome biogenesis, these results indicate that rRNA accumulation and ribosome biogenesis surprisingly restrict HCMV replication. Moreover, it suggests that ribosome biogenesis comprises part of a cell intrinsic immune response that limits virus reproduction.

## Suppressing ribosome biogenesis stimulates HCMV reproduction by restricting type I interferon production

A major mechanism through which innate, cell-intrinsic responses limit virus replication is through the products of interferon-stimulated genes (ISGs) (*Schneider et al., 2014*). The possibility that ribosome biogenesis might impact innate immune responses in HCMV-infected primary fibroblasts and interfering with ribosome biogenesis might impair ISG expression was therefore considered. Compared to HCMV-infected cultures treated with non-silencing siRNA, reduced levels of representative ISG-encoded mRNAs (*Figure 4A*) and proteins (*Figure 4B*) accumulated in cultures depleted for either of the RNAPI transcription factors TIF-IA or UBF, the 5S RNA RNAPIII transcription factor TFIIIA, or the rRNA processing factor BOP1. Accordingly, representative immediate early (IE), early (E) and Late (L) viral proteins accumulated to greater levels in cultures where ribosome biogenesis was suppressed by either TIF-IA, UBF, TFIIIA, or BOP1 siRNA as opposed to non-silencing siRNA (*Figure 4C*). To determine whether reduced HCMV IE, E and L protein accumulation in infected cells exposed to non-silencing siRNA compared to TIF-IA, UBF, TFIIIA, or BOP1 siRNA was dependent upon type I IFN action, cells were treated with a Janus kinase inhibitor (JAKi). By inhibiting JAK and preventing IFN signaling through its cell surface receptor, JAKi suppresses ISG expression. In contrast to DMSO-treated cells where viral protein abundance was far less in cells exposed to non-silencing siRNA compared to siRNAs that inhibited ribosome biogenesis, viral protein levels in JAKi-treated cells exposed to non-silencing siRNA were approximately equivalent to those detected in TIF-IA, UBF, TFIIIA, or BOP1 siRNA-treated NHDFs (*Figure 4C*). This demonstrates that IFN signaling via JAK, which normally triggers ISG expression and subsequently limits viral protein accumulation, can be overcome by interfering with ribosome biogenesis. In addition, it is consistent with the notion that ribosome biogenesis in HCMV-infected cells is required to establish an anti-viral state dependent upon ISG expression that limits virus replication. Finally, it further suggests that inhibiting ribosome biogenesis by interfering with TIF-IA, UBF, TFIIIA or BOP1 limited JAK-dependent signaling perhaps by modulating type I interferon production.

Having demonstrated that interfering with ribosome biogenesis restricts ISG expression, we next tested whether rRNA synthesis impacted the capacity of infected cells to produce IFNB1 mRNA. NHDFs treated with non-silencing siRNA or siRNAs specific for TIF-IA were infected with HCMV and the accumulation of IFNB1 mRNA measured at six hpi. Surprisingly, accumulation of IFNB1 mRNA and a representative ISG-encoded polypeptide (IFIT2 protein) were significantly reduced in response to HCMV infection when cells were treated with TIF-IA siRNA compared to non-silencing siRNA (*Figure 4D,E*). Representative ISG mRNAs accumulated to similar levels following exposure of cells treated with TIF-IA siRNA or non-silencing siRNA to recombinant type I IFN (*Figure 4—figure*

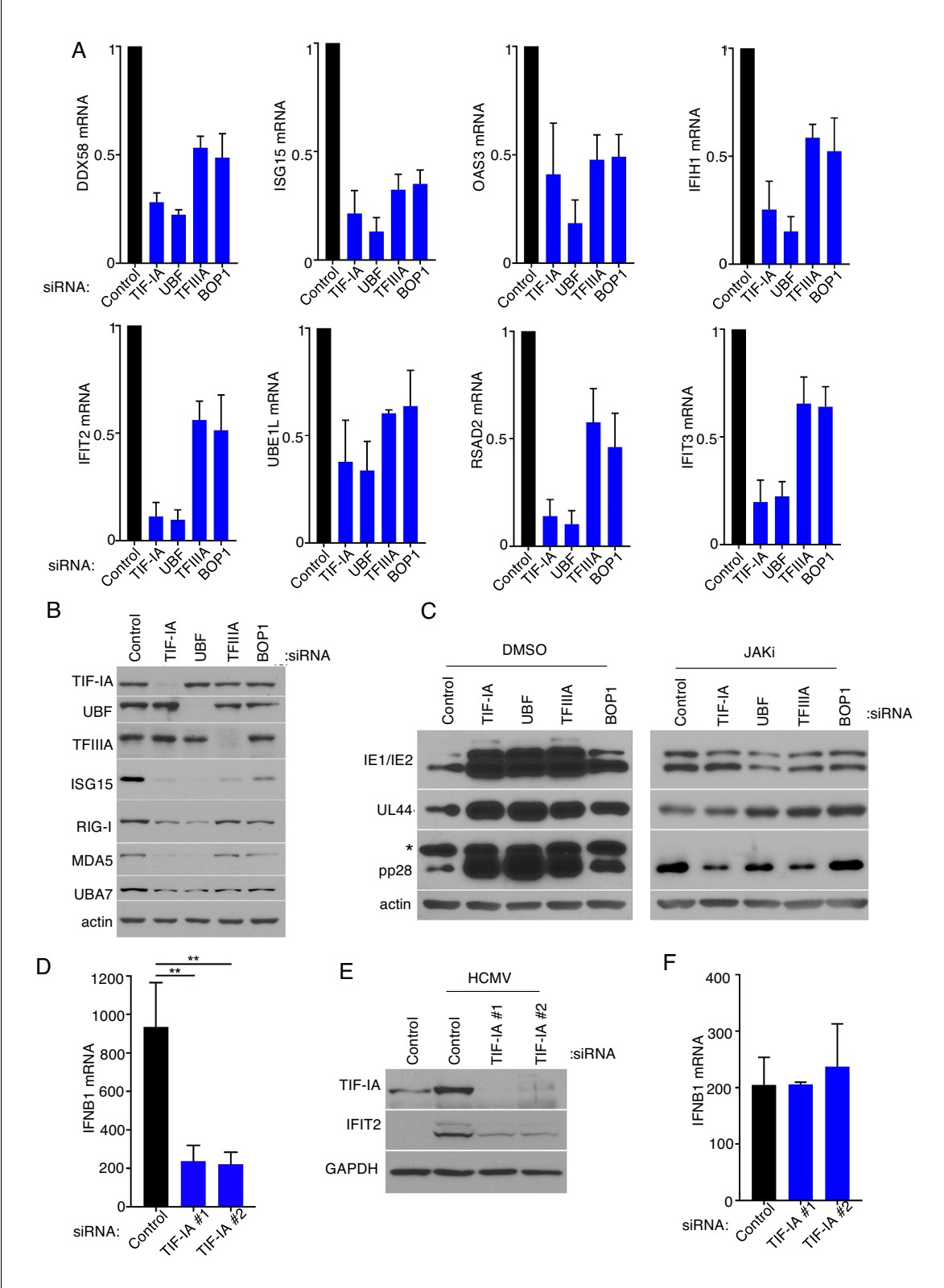

**Figure 4.** Ribosome biogenesis impairs HCMV protein accumulation by augmenting the innate immune reponse. (a) NHDFs were treated with non-silencing (ns) control, TIF-IA, UBF, TFIIIA, or BOP1 siRNAs and infected with HCMV (MOI = 3 PFU/cell) two dpi total RNA was isolated and RT-qPCR analysis was performed for DDX58, IFIT2, ISG15, IFIT3, RSAD2, OAS3, UBA7, and IFIH1 mRNA (n = 3). (b) As in (a) except total protein was collected, fractionated by SDS-PAGE, and analyzed by immunoblotting using antibodies specific for TIF-IA, UBF, TFIIIA, ISG15, RIG-I, MDA5, UBA7, and actin

*Figure 4 continued on next page*

*Figure 4 continued*

(loading control) (n = 2). (c) NHDFs were transfected with the indicated siRNAs. 3 d post-transfection cells were treated with DMSO or JAKi (10 µM) prior to infection with HCMV (MOI = 0.05 pfu/cell) 5 d post-infection total protein was collected, and immunoblotting was performed for IE1/IE2, UL44, pp28, and actin (loading control). *Denotes a non-specific band frequently observed on overexposed blots using anti-pp28 (n = 2). (d,e) NHDFs were transfected with ns control or TIF-IA siRNAs and infected with HCMV(MOI = 3 pfu/cell). (d) Six hpi total RNA was isolated and RT-qPCR was performed using primers specific for IFNB1 mRNA. The error bars indicate SEM. **p≤0.01; Student's *t* test (n = 3). (e) 48 hpi total protein was collected, fractionated by SDS- PAGE, and analyzed by immunoblotting using antibodies specific for TIF-IA, IFIT2, and GAPDH (loading control) (f). As in (d) except cells were infected with VSV (MOI = 3 PFU/Cell).

The online version of this article includes the following figure supplement(s) for figure 4:

**Figure supplement 1.** Depletion of RNA polymerase I factors does not impact interferon signaling.

*supplement 1*), excluding the possibility that the decrease in ISGs resulted from an impaired response to type-I-interferon. As depleting TIF-IA had no detectable impact on IFNB1 mRNA accumulation in response to infection with the RNA virus vesicular stomatitis virus (VSV), this suggested that rRNA biogenesis selectively influenced IFNB1 induction in response to viruses like HCMV with dsDNA genomes (*Figure 4F*) (*Yanai et al., 2009*). Moreover, it raised the possibility that the regulation of IFNB1 production by ribosome biogenesis in HCMV infected cells might not require viral gene expression and instead reflect cell intrinsic responses to dsDNA, a pathogen-associated molecule that engages host pattern recognition receptors.

## Induction of interferon-beta mRNA in response to dsDNA in uninfected primary fibroblasts is impaired by depleting TIF-IA

To investigate how ribosome biogenesis might restrict HCMV replication, the possibility that rRNA production might be linked to a cell intrinsic host defense was investigated. To address this possibility, 45S rRNA levels in mock-infected and NHDFs infected with UV-inactivated HCMV were measured by qPCR and compared. *Figure 5A* shows that UV-inactivated HCMV stimulated 45S rRNA accumulation, which suggested that viral gene expression was not required to induce rRNA accumulation and is consistent with rRNA accumulation being triggered in response to either a virion protein component or a pathogen- associated molecular pattern like a nucleic acid. Since HCMV is a DNA virus, how DNA might impact 45S rRNA accumulation in uninfected NHDFs was investigated. Indeed, 45S rRNA abundance increased following treatment with two different immunostimulatory synthetic dsDNAs, but not the same single-stranded DNA sequences (*Figure 5B*). Similarly, circular plasmid dsDNA from bacteria stimulated 45S rRNA accumulation (*Figure 5B*). Thus, 45S rRNA accumulation in uninfected NHDFs increased in response to ds, but not single-stranded DNA irrespective of whether the DNA was linear or circular.

Because dsDNA is the major PAMP delivered by HCMV, and TIF-IA-depletion did not detectably suppress IFN induction by an RNA virus, the possibility that rRNA biogenesis might regulate IFN induction in uninfected cells was evaluated. NHDFs treated with non-silencing siRNA or siRNAs specific for TIF-IA were transfected with a synthetic immunostimulatory dsDNA and levels of representative ISG-encoded proteins and IFNB1 mRNA measured. Compared with non-silencing siRNA-treated cells, accumulation of IFNB1 mRNA and two representative ISG-encoded proteins, USP18 and IFIT2, was reduced in TIF-IA-depleted cells exposed to dsDNA (*Figure 5C,D*). Next, the possibility that this phenotype was not due specifically to loss of TIF-IA, but instead due to impairment of RNAPI, was addressed by depleting UBF, a critical RNAPI transcription factor. Cells treated with ns siRNA, TIF-IA siRNA, or UBF siRNA were transfected with dsDNA. Similar to observations in TIF-IA depleted cells, IFNB1 mRNA accumulated to reduced levels in cells treated with UBF siRNA compared with non-silencing siRNA (*Figure 5—figure supplement 1*). This demonstrates that IFNB1 mRNA induction by dsDNA in uninfected primary human fibroblasts is regulated by RNAPI transcription factors.

RNA-seq was performed on poly(A) RNA to evaluate genome-wide transcriptomic changes in dsDNA transfected cells that were specifically responsive to the RNAPI transcription factors TIF-IA and UBF. As expected, gene ontology analysis revealed an enrichment of genes involved in response to viral infection, interferon signaling, and the innate immune response in the set of genes which were upregulated more than four-fold in response to synthetic immunostimulatory dsDNA in ns siRNA-treated cells (*Figure 5—figure supplement 2*). Strikingly, induction of all the genes most highly-induced (>4 fold) by dsDNA was reduced by depleting either TIF-IA or UBF (*Figure 5E*;

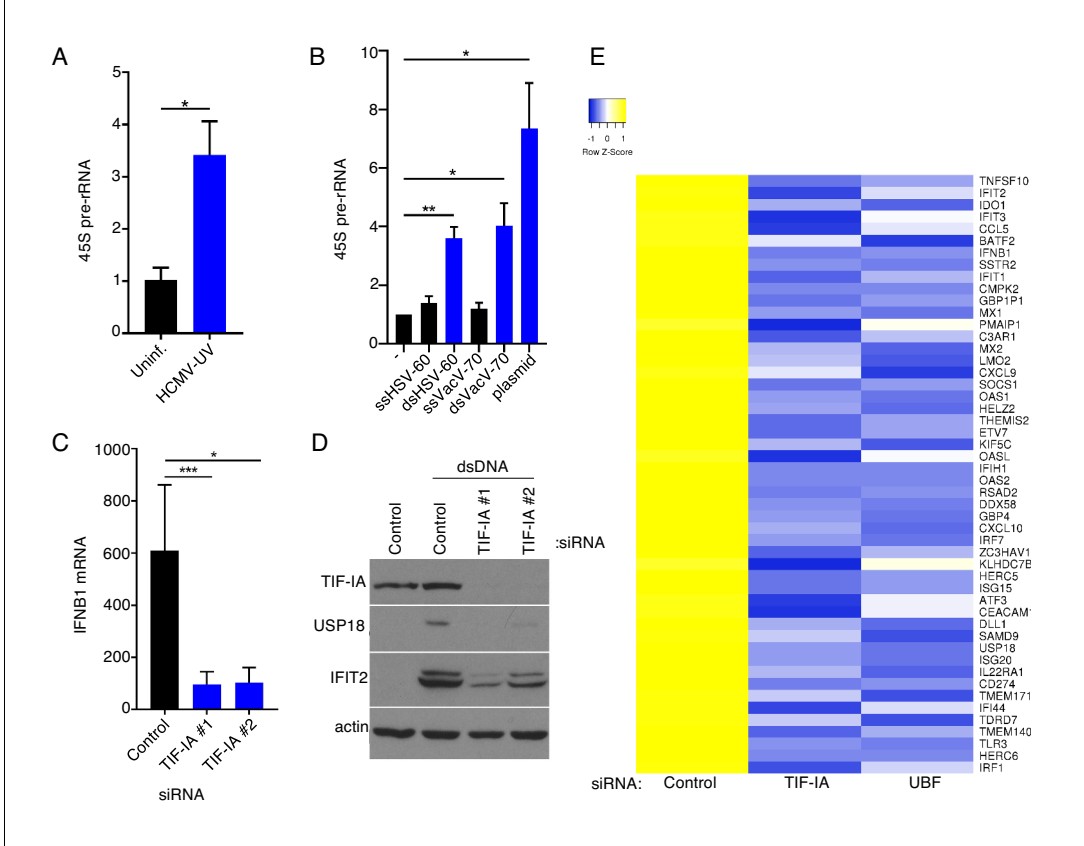

**Figure 5.** RNA polymerase I factors control accumulation of dsDNA-responsive genes. (**a**) Growth arrested NHDFs were uninfected or infected with UV-inactivated HCMV (MOI = 3 PFU/Cell) 24 hr post-transfection total RNA was isolated and RT- qPCR analysis was performed for 45S pre-rRNA. The error bars indicate SEM. *p≤0.05; Student's *t* test (n = 3). (**b**) Growth arrested NHDFs were transfected with the indicated DNA. 24 hr post-transfection total RNA was isolated and RT-qPCR analysis was performed for 45S pre-rRNA. Error bars indicate SEM. *p≤0.05; **p≤0.01; Student's *t* test (n = 4). (**c**) NHDFs were transfected with non-silencing (ns) control or TIF-IA specific siRNAs. After 3 d, cultures were transfected with either no DNA or dsVacV-70 (dsDNA.) 6 hr post-transfection total RNA was isolated and RT-qPCR was performed using primers specific for IFNB1 mRNA. (n = 3) Error bars indicate SEM. *p≤0.05; ***p≤0.001; Student's *t* test. (**d**) As in (**c**), except 24 hr post-transfection total protein was collected, fractionated by SDS-PAGE, and analyzed by immunoblotting using antibodies specific for TIF-IA, USP18, IFIT2, and actin (loading control). (**e**) NHDFs were transfected with ns, TIF-IA specific, or UBF specific siRNA. After 3 d, cells were transfected with no DNA or dsDNA for 6 hr after which total RNA was isolated and poly(A) selected. RNA-seq was performed and a heatmap showing differential regulation in TIF-IA and UBF siRNA treated cells of the top 50 genes induced by dsDNA in ns siRNA treated cells was generated.

The online version of this article includes the following figure supplement(s) for figure 5:

**Figure supplement 1.** UBF is required for efficient dsDNA-induced accumulation of IFNB1 mRNA.

**Figure supplement 2.** Genome-Wide responses to dsDNA.

*Supplementary file 1*). This suggests that mounting an effective cell intrinsic immune response induced by dsDNA-sensing is reliant on RNAPI transcription factors and consistent with a role for RNAPI activity in this process. Moreover, it establishes a fundamental connection between rRNA and ribosome biogenesis and innate immune responses in uninfected primary fibroblasts.

## TIF-IA depletion regulates expression of NF-Y and p53-responsive DREAM complex target genes including the DNA sensor HMGB2

To determine how RNAPI transcription might regulate innate responses to dsDNA, the impact of interfering with RNAPI transcription on global gene expression was analyzed by RNA-seq. Following normalization and classification of differentially regulated genes (adjusted P-value [Padj] of <0.01), 3135 genes proved responsive to TIF-IA depletion compared to non-silencing siRNA control-treated cultures. While 1392 genes were downregulated following TIF-IA- depletion, 1743 genes were upregulated (*Figure 6A*; *Supplementary file 2*). Gene ontology analysis showed that compared to non-

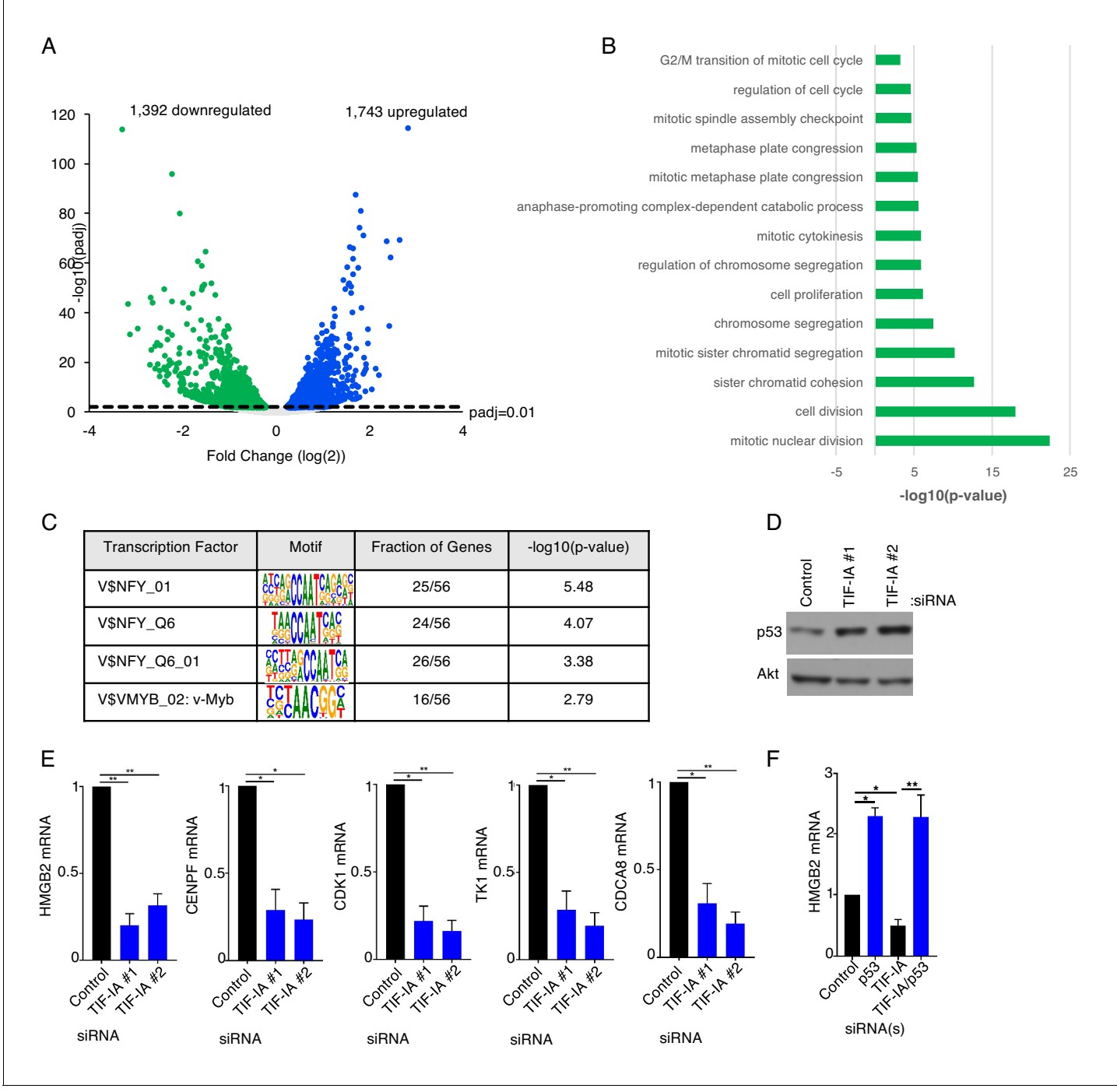

**Figure 6.** TIF-IA controls the abundances of nuclear factor Y and DREAM complex transcriptional targets. (**a,b**) NHDFs were transfected with non-silencing (ns) control siRNA or a TIF-IA siRNA. Three days post transfection total RNA was isolated and RNA-seq was performed and (**a**) a volcano plot showing differentially expressed genes identified from RNA-seq of cells treated with TIF-IA siRNA (adjusted p-value<0.01) was generated and (**b**) Gene ontology analysis (GOTERM_BP_DIRECT) of genes downregulated more than four-fold in TIF-IA depleted cells compared to ns siRNA treated cells was performed using DAVID Functional Annotation Clustering Tool (n = 2). (**c**) TRANSFAC analysis of genes downregulated more than log(2) in TIF-IA depleted cells compared to ns siRNA treated cells was performed using GATHER to identify common modes of transcriptional regulation. (**d**) 3 d post-transfection of NHDFs with ns or TIF-IA targeting siRNAs, total protein was collected, separated by SDS-PAGE, and analyzed by immunoblotting using antibodies specific for p53 and Akt (loading control) (n = 3). (**e**) 3 d post-transfection of NHDFs with ns or TIF-IA targeting siRNAs, total RNA was isolated and RT-qPCR was performed for HMGB2 mRNA, CENPF mRNA, CDK1 mRNA, TK1 mRNA and CDCA8 mRNA. Error bars indicate SEM. *p≤0.05; **p≤0.01; Student's *t* test (n = 3). (**f**) As in d using the indicated siRNAs and measuring HMGB2 mRNA levels.

silencing (ns) control siRNA treated cells, genes in pathways controlling mitotic nuclear division (p<3.90E-23), cell division (p<9.12E-19), sister chromatid cohesion (p<2.07E-13), cell proliferation (p<6.60E-07), regulation of cell cycle (p<2.42E-05) and G2/M transition of mitotic cell cycle (p<5.76E-04) were down regulated in TIF-IA depleted cells (*Figure 6B*), consistent with the role for ribosome biogenesis in cellular growth and proliferation (*Donati et al., 2012*).

To investigate whether a subset of genes responsive to TIF-IA-depletion might be co-regulated by a discrete transcriptional program, binding sites for transcription factors were interrogated using the Transfac database on eukaryotic transcriptional regulation, which contains data on transcription factors, their target genes, and regulatory binding sites (*Matys et al., 2003*). Results of this analysis revealed that expression of genes responsive to the transcription factor Nuclear Factor-Y (NF-Y) were found to be selectively repressed by TIF-IA-depletion (*Figure 6C*). Furthermore, NF-Y target genes reportedly overlap significantly ($p=2.2\times10^{-16}$) with DREAM complex targets (*Sadasivam et al., 2012*), and many DREAM target genes encode cell cycle genes indirectly repressed by p53 (*Fischer et al., 2016*). Interfering with ribosome biogenesis also results in nucleolar stress, which stabilizes p53 (*Boulon et al., 2010*; *Chakraborty et al., 2011*; *Golomb et al., 2014*; *Deisenroth et al., 2016*). Indeed, p53 accumulated in response to TIF-IA depletion (*Figure 6D*) in agreement with published studies (*Yuan et al., 2005*) and consistent with our observed changes in p53 target gene expression. Measuring gene expression by qPCR following TIF-IA depletion demonstrated that levels for five different DREAM target genes [HMGB2, CENPF, CDK1, TK1, and CDCA8] decreased, effectively validating the results of our genome-wide analysis (*Figure 6E*). HMGB2 figured prominently among this set of genes as HMGB2-deficient murine cells have been previously shown to be defective for type I interferon and inflammatory cytokine induction in response to dsDNA and HMGB2 has been proposed to play a critical role in cytoplasmic dsDNA signaling (*Lee et al., 2013*; *Yanai et al., 2009*). Moreover, the reduction of HMGB2 mRNA in response to TIF-IA depletion was p53-dependent as measured by qRT-PCR (*Figure 6F*), in accord with prior genome-wide data (*Fischer et al., 2016*). Given the established role of HMGB2 in type I interferon production in mice (*Yanai et al., 2009*; *Yanai et al., 2012*), the accumulation of p53 in response to TIF-IA-depletion (*Figure 6D*; *Yuan et al., 2005*), and repression of HMGB2 gene expression by p53 (*Figure 6F*), the impact of interfering with TIF-IA abundance upon HMGB2 levels was next investigated.

## Regulation of HMGB2 abundance by TIF-IA

HMGB2 is a non-specific dsDNA binding protein that is loosely associated with chromatin (*Malarkey and Churchill, 2012*). In addition, HMGB proteins are alarmins and upon release from injured cells are capable of activating immune cells and evoking inflammation (*Harris et al., 2012*; *Yanai et al., 2012*). Nuclear retention of HMGB family members limits inflammation (*Avgousti et al., 2016*). To further investigate how TIF-IA might impact HMGB2, the overall abundance of the HMGB2 polypeptide and its subcellular distribution were evaluated in cells treated with non-silencing siRNA vs. two different TIF-IA siRNAs. Compared to NHDFs treated with non-silencing siRNA, HMGB2 abundance was reduced in TIF-IA siRNA-treated cells whereas actin levels remained relatively constant. (*Figure 7A*). A similar reduction in HMGB2 mRNA and protein levels was observed in NHDFs treated with the RNA pol I selective inhibitor CX-5461 (*Figure 7—figure supplement 1*). In addition, the reduction in HMGB2 mRNA abundance in response to CX-5461 was dependent upon p53 (*Figure 7—figure supplement 1*). This shows that TIF-IA depletion or an RNA pol I selective chemical inhibitor similarly reduced HMGB2 levels, and is consistent with TIF-IA-depletion influencing HMGB2 abundance by inhibiting RNA pol I transcription. In NHDFs treated with non-silencing siRNA, HMGB2 was detected throughout the cell in both cytoplasmic and nuclear compartments. However, although nuclear HMGB2 was readily detected in TIF-IA depleted NHDFs, cytoplasmic HMGB2 staining was unexpectedly lost (*Figure 7B*). Importantly, HMGB2 detection in both nuclear and cytoplasmic compartments was equally abrogated upon HMGB2-depletion using RNAi, validating that the antibody signal obtained via immuno-cytochemistry was specific for HMGB2 antigen (*Figure 7B*). Taken together, this demonstrates that interfering with ribosome biogenesis via TIF-IA depletion reduced HMGB2 mRNA levels and selectively reduced HMGB2 protein abundance in the cytoplasm. As IFNB1 mRNA induction can be triggered by dsDNA-sensing in the cytoplasm, the selective loss of cytoplasmic HMGB2 following TIF-IA-depletion potentially limits the cellular capacity to sense dsDNA.

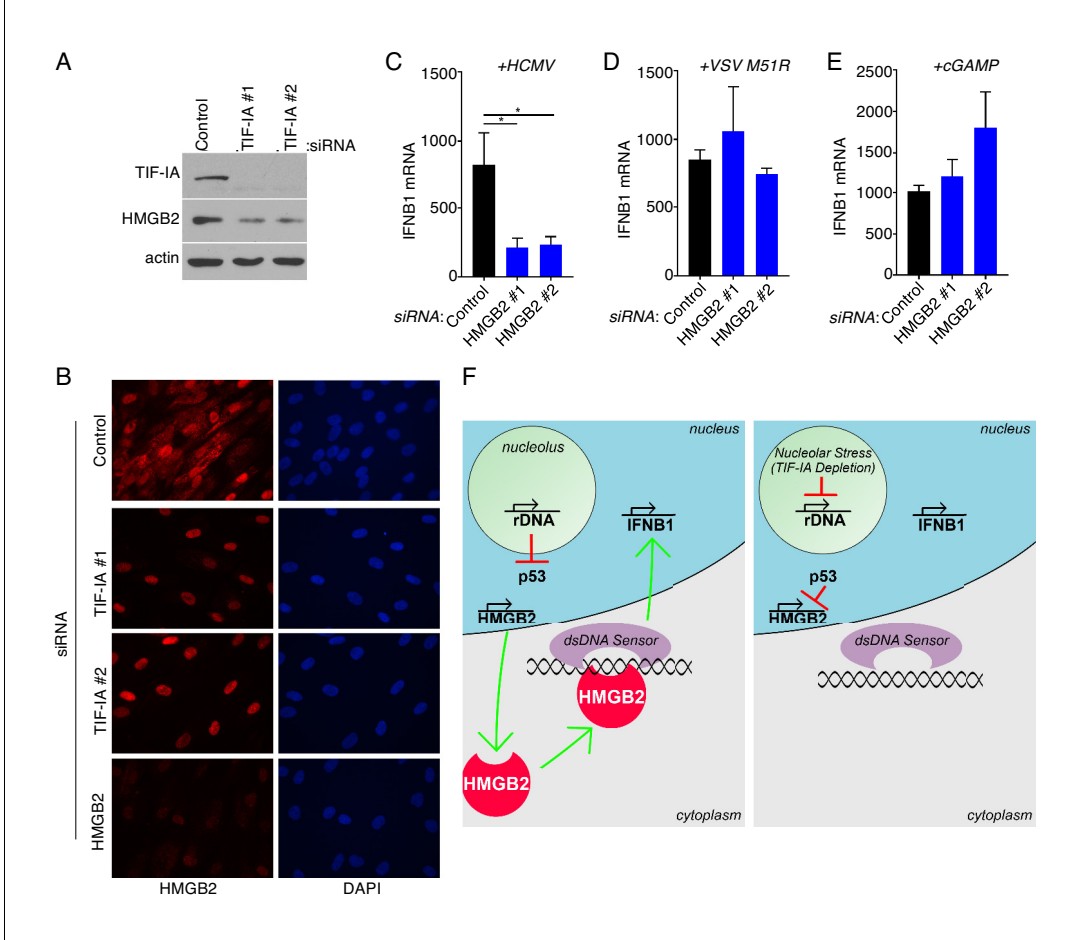

**Figure 7.** Cytoplasmic HMGB2 abundance is controlled by TIF-IA and dictates the cellular responses to dsDNA. (a) 3 d post-transfection of NHDFs with non-silencing (ns) control or TIF-IA targeting siRNAs, protein was collected, separated on an SDS-PAGE gel, and analyzed by immunoblotting using antibodies specific for TIF-IA, HMGB2, and actin (loading control) (n = 2). (b) NHDFs were transfected with ns, TIF-IA, or HMGB2 siRNAs. 3 d post-transfection cells were fixed in 4% PFA and immunofluorescence staining was performed using an antibody specific for HMGB2 (n = 2). Nuclei were visualized following DAPI staining. (c–e) NHDFs were transfected with ns or HMGB2 siRNA and (c) infected with HCMV, (d) infected with VSV, or (e) treated with cGAMP and (c–e) 6 hr later, total RNA was isolated, and RT- qPCR was performed using primers specific for IFNB1 mRNA. Error bars indicate SEM. *$p \leq 0.05$; Student's $t$ test (n = 3). (f) model depicting how ribosome biogenesis regulates cellular responses to dsDNA via HMGB2. (*Left panel*) Normal rDNA transcription precludes nucleolar stress-induced p53 stabilization, permitting HMGB2 transcription by RNA pol II in the nucleoplasm and HMGB2 protein (shown in red) accumulation in the cytoplasm. Upon detecting non-microbial or HCMV cytoplasmic dsDNA, HMGB2 either alone or together with a dsDNA sensor like cGAS stimulates IFNB1 mRNA accumulation. (*Right panel*) Inhibiting RNAPI by TIF-IA depletion results in nucleolar stress and stabilizes p53, which in turn represses HMGB2 transcription, reduces HMGB2 protein abundance and restricts IFNB1 transcription in response to dsDNA.

The online version of this article includes the following figure supplement(s) for figure 7:

**Figure supplement 1.** Inhibiting RNA polymerase I reduces HMGB2 abundance in a p53-dependent manner.

## Interferon induction in response to HCMV requires HMGB2

Having established that interfering with ribosome biogenesis stimulated HCMV productive growth by limiting IFNB1 induction, and reduced cytoplasmic HMGB2 levels in uninfected cells, a role for HMGB2 in regulating IFNB1 induction in response to HCMV was next considered. To interrogate the requirement for HMGB2 in dsDNA induced IFNB1 mRNA accumulation in human cells, HMGB2 was depleted from NHDFs prior to infection with representative DNA and RNA viruses. Compared to non-silencing (ns) control siRNA treated cultures, significantly less IFNB1 mRNA accumulated in response to HCMV infection upon HMGB2-depletion (*Figure 7C*). By contrast, HMGB2-depletion had no detectable impact on IFNB1 accumulation induced in response to infection with VSV, an RNA virus (*Figure 7D*), in agreement with reports showing HMGB2 was dispensable for IFNB1

induction by RNA in murine cells (*Yanai et al., 2009*). To further demonstrate that HMGB2 acts at the level of DNA sensing, dsDNA sensing was bypassed by treating cells with cGAMP, the second messenger produced by cGAS that directly activates STING (*Wu et al., 2013*). Compared to ns siRNA-treated cultures, IFNB1 mRNA accumulation in response to cGAMP was not detectably reduced by HMGB2-depletion (*Figure 7E*). That HMGB2-depletion did not detectably impair IFNB1 induction by the RNA virus VSV or cGAMP suggests that HMGB2 is selectively required for sensing dsDNA, as opposed to RNA, upstream of STING in human cells. Taken together, these results establish a critical role for HMGB2, whose levels decrease in response to TIF-IA-depletion, in the induction of IFNB1 by HCMV.

## Discussion

Ribosome biogenesis is critical for normal cell proliferation and highly responsive to metabolic, physiological and environmental challenges. However, the impact of ribosome biogenesis on virus infection biology and whether it might be linked to innate immune responses is poorly understood. Given the universal dependence of virus protein synthesis upon cellular ribosomes, it is assumed that ribosome biogenesis would support ongoing protein synthesis and virus reproduction. Here, we show that while interfering with ribosome biogenesis suppressed ongoing protein synthesis in uninfected primary fibroblasts, it did not detectably impact protein synthesis in HCMV-infected cells. By contrast, reducing ribosome biogenesis unexpectedly stimulated HCMV reproduction. Genome-wide transcriptome analysis revealed that interfering with rRNA accumulation decreased expression of genes involved in cell division, many of which were regulated by NF-Y or the DREAM complex. Among these was High Mobility Group Box 2 (HMGB2), a p53-repressed gene whose product is found both bound to chromatin and in the cytoplasm where it is likely critical for innate immune responses to dsDNA. Significantly, depleting the RNAPI specific transcription factor TIF-IA impaired IFNB1 mRNA accumulation in response to HCMV or dsDNA. Thus, preventing rRNA accumulation dampens cellular responses to dsDNA in part by regulating levels of the DNA sensor HMGB2 (*Figure 7F*). Moreover, it suggests that rRNA biogenesis triggers an HMGB2-dsDNA sensing pathway and reveals a surprising role for rRNA accumulation and/or nucleolar activity in cell intrinsic innate immunity.

Our results show that rRNA accumulation, ribosome biogenesis and DNA sensing can be integrated to coordinate IFNB1 production. Inhibiting ribosome biogenesis triggers nucleolar stress, which is known to stabilize p53 (*Boulon et al., 2010*; *Chakraborty et al., 2011*; *Golomb et al., 2014*; *Deisenroth et al., 2016*) and likely accounts for our observed p53-dependent repression of HMGB2 expression. This allows HMGB2 levels to be directly tuned to nucleolar stress and dampened accordingly to limit IFNB1 transcription in response to dsDNA sensing (*Figure 7F*). Further investigation might reveal that additional genes whose expression is regulated upon TIF-IA-depletion might also contribute to proper control of cell intrinsic innate immune responses. While RNAP III, which transcribes nuclear genes encoding 5S rRNA and tRNAs, reportedly transcribes cytoplasmic dsDNA into a RIG-I RNA substrate (*Chiu et al., 2009*), nucleolar or ribosome biogenesis functions of the enzymes were not implicated previously. Additional indirect evidence consistent with a role for ribosomes in controlling innate immunity, however, is mounting. Formation of a multipartite complex including STING and S6K is needed to activate IRF3 in response to DNA sensing (*Wang et al., 2016*) A ribosomal protein deficiency resulting in Diamond-Blackfan Anemia (DBA) impairs rRNA processing and ribosome biogenesis, resulting in activation of innate immune signaling (*Danilova et al., 2018*). The enigmatic cleavage of UBF by poliovirus protease 3C(pro), which inhibits pol I transcription (*Banerjee et al., 2005*), can now be considered a strategy to limit IFNB1 production. In addition, antagonizing RNAPI ameliorates experimental autoimmune encephalomyelitis in animal models and is associated with a benign course of multiple sclerosis (*Achiron et al., 2013*).

Deregulated ribosome biogenesis is associated with human diseases that may be differentially impacted by IFNB1 production. For example, proliferative diseases like cancer are associated with hyperactivated rRNA synthesis and ribosome biogenesis. In normal cells, IFNB1 produced in response to ribosome biogenesis may limit proliferation and serve to restrict tumor development. This could in part account for why many tumors have impaired nucleic acid sensing and/or IFN pathways (*Critchley-Thorne et al., 2009*; *Vanpouille-Box et al., 2018*). Hutchinson-Gilford progeria

syndrome (HGPS) is a premature aging disease characterized by expression of a truncated form of Lamin A called progerin, nucleolar expansion, enhanced ribosome biogenesis, and elevated interferon production (*Buchwalter and Hetzer, 2017*). The correlation between ribosome biogenesis and ISG product accumulation raises the possibility that HGPS may be considered an interferonopathy precipitated by pathogenic enhancement of ribosome production. In contrast, ribosomopathies result from decreased ribosome biogenesis. Patients with Shwachman-Bodian-Diamond Syndrome (SBDS), an inherited ribosomopathy characterized by pancreatic, bone marrow, and immune dysfunction (*Giri et al., 2015*), experience elevated rates of infectious complications, such as those involving *Pneumococcus, Staphylococcus*, and Parvovirus B19 (*Grinspan and Pikora, 2005*; *Miniero et al., 1996*). Poor innate immune responses, including reduced IFN production, resulting from decreased ribosome biogenesis may be a contributing factor.

While an HMGB2-dependent DNA-sensing pathway links ribosome biogenesis and IFN production, the identity of the non-microbial DNA detected remains enigmatic. Increases in rDNA instability caused by nucleolar dysfunction could trigger a DDR, which reportedly can activate cGAS to stimulate increased interferon production (*Kobayashi, 2008*). HMGB2 facilitates sensing of long dsDNA by cGAS (*Andreeva et al., 2017*). Accordingly, decreased RNAPI-induced DNA damage could in part explain why inhibiting ribosome biogenesis limits innate immune responses. However, impairing rRNA accumulation by dampening rRNA synthesis also reportedly can activate DDR pathways (*Calo et al., 2018*). Further investigation is needed to resolve this paradox. Nevertheless, failsafe pathways likely detect cytoplasmic dsDNA resulting from rDNA transcription activation, much like cGAS detects cytoplasmic dsDNA during senescence.

In addition to IFNB1 production, both cGAS and HMGB2 regulate senescence and HMGB2 plays a role controlling the senescence-associated secretory pathway (*Aird et al., 2016*; *Guerrero and Gil, 2016*; *Yang et al., 2017*; *Zirkel et al., 2018*). Activation of dsDNA sensing pathways by ribosome biogenesis might contribute to the reported associations between ribosome biogenesis, lifespan and aging (*Sharifi and Bierhoff, 2018*; *Tiku and Antebi, 2018*).For example, by activating cytoplasmic dsDNA-sensing, rDNA instability resulting from elevated ribosome biogenesis observed in HGPS could accelerate senescence and thereby drive the progression and severity of premature aging.

Our findings challenge notions that ribosome biogenesis is universally required to support sustained mRNA translation. While corroborating evidence was found in uninfected primary human fibroblasts, it was not the case in HCMV-infected cells. Although polysome levels and protein synthesis were not detectably altered by TIF-IA-depletion in HCMV-infected fibroblasts, precisely why infected compared to uninfected cell protein synthesis was insensitive to interfering with ribosome biogenesis remains unknown. The underlying mechanism(s) are potentially multifaceted and complex perhaps involving virus-induced increases in cellular translation factors that allow more efficient use of existing ribosomes (*McKinney et al., 2012*; *McKinney et al., 2014*), differences among ribosome populations, cis-acting elements (*Mizrahi et al., 2018*) in host and viral mRNAs being translated, specific virus-encoded polypeptides, and/or the differential complexity of mRNA populations in uninfected compared to HCMV-infected cells. Changes in rRNA transcription also reportedly influence proliferation and stem cell fate (*Zhang et al., 2014*). Notably, sustained protein synthesis despite reduced nucleolar volume and rRNA synthesis has been observed in germline stem cell differentiation in *Drosophila* (*Neumüller et al., 2008*; *Sanchez et al., 2016*). Likewise, rRNA synthesis is not required for growth factor-mediated hypertrophy of human primary myotubes (*Crossland et al., 2017*). Thus, rRNA and ribosome biogenesis can be uncoupled from protein synthesis in invertebrates and in primary human cells. Further study is required to understand how rRNA and ribosome biogenesis impinges on these different downstream effectors, including HMGB2 protein and mRNA levels. Given the large energy costs associated with ribosome building, it is perhaps no wonder that it is integrated into vital cellular responses that govern survival, including proliferation, lifespan and now cell intrinsic immunity.

# Materials and methods

## Cells and viruses

Normal Human Dermal Fibroblasts (NHDFs) and HCMV AD169GFP propagation and quantifiaction have been previously described (*Bianco and Mohr, 2017*). VSV-M51R-GFP (kindly provided by the laboratory of Benjamin TenOever, Mt. Sinai School of Medicine, NY) was propagated in Vero cells and quantified by a standard plaque assay.

## RT-qPCR

RNA was isolated from cells using TRIzol Reagent (15596026; ThermoFisher Scientific) as previously described (*Bianco and Mohr, 2017*). cDNA was synthesized from 250 ng of purified RNA using qScript XLT Supermix (Quanta; 95161) according to the manufacturer's instructions. qPCR was performed in a Bio-Rad CFX96 RT-qPCR instrument using SsoAdvanced Universal SYBR Green Supermix (1725274; Bio-Rad) and the primers described in *Supplementary file 3* with an annealing temperature of 58°C.

## DNA and DNA transfections

DNA transfections were performed as previously described (*Bianco and Mohr, 2017*). VacV-70 oligomers (5'-CCATCAGAAAGAGGTTTAATATTTTTGTGAGACCATCGAAGAGAGAAAGAGA TAAAACTTTTTTACGACT-3' and 5'- AGTCGTAAAAAAGTTTTATCTCTTTCTCTCTTCGATGGTC TCACAAAAATATTAAACCTCTTTCTGATGG-3') were synthesized by Integrated DNA Technologies and annealed as previously described (*Bianco and Mohr, 2017*). The RNAPI reporter plasmid (pHrP2-BH) was a gift from Ingrid Grummt (*Mayer et al., 2005*).

## Immunoblotting and antibodies

Cells were lysed, and immunoblotting was performed as previously described (*Bianco and Mohr, 2017*). The antibodies used in this study were as follows: TIF-IA (Bethyl Laboratories; A303-065A) UBF (Santa Cruz; SC-13125), Akt (Cell Signaling; 9272), Fibrillarin (Santa Cruz; sc-166001), UL44 (Virusys; CA006), pp28 (Virusys; CA004-100), IE1/IE2 (Millipore; MAB810), HMGB2 (Cell Signaling Technology; 14163), USP18 (Cell Signaling Technology; 4813), IFIT2 (Proteintech; 12604–1-AP), TFIIIA (Bethyl; A303-621A), ISG15 (Proteintech; 15981–1-AP), RIG-I (Proteintech; 20566), MDA5 (Proteintech; 21775–1-AP), UBA7 (Cell Signaling Technology; 69023), actin (Cell Signaling Technology; 3700), BrdU (Sigma; B2531), p53 (Cell Signaling Technology; 9282).

## Polysome analysis

$10^7$ NHDF cells were incubated with 100 μg/mL cycloheximide (Sigma; C7698) for 10 min. at 37°C 5% $CO_2$ prior to lysis in polysome lysis buffer (15 mM Tris (pH 7.5), 0.3M NaCl, 15 mM $MgCl_2$, 100 μg/mL cycloheximide) containing 100 U/mL RiboLock RNase Inhibitor (ThermoFisher Scientific; EO0381.) After incubation on ice for 10 min., nuclei and mitochondria were pelleted by centrifugation at 14,000 RPM for 5 min. at 4°C. Cytoplasmic lysate was layered onto 10–50% sucrose gradients (in polysome buffer with 100 μg/mL cycloheximide) in thinwall polypropylene ultracentrifuge tubes (Beckman Coulter; 331372). Gradients were then spun at 38,000 RPM for 2 hr in a SW41Ti rotor (Beckman Coulter; 331362) at 4°C. Absorbance profiles were produced by pumping the gradients through a flow cell while measuring the absorbance of RNA at 254 nM using a Density Gradient Fractionation System (Brandel; BR-188).

## Immunofluorescence and metabolic labeling with [$^{35}$S]-amino acids and 5-fluorouridine

Immunofluorescence and [$^{35}$S]-amino acid labeling were performed as previously described (*Burgess and Mohr, 2015*). To label nascent rRNA, cells were incubated with 2 mM 5-fluorouridine (Sigma; F5130-100MG) for 20 min. prior to fixation.

## RNA sequencing

Total RNA was isolated and DNAse I treated on RNeasy columns (74106; Qiagen). Illumina Truseq stranded poly(A)-selected mRNA libraries were prepared, multiplexed and sequenced in single-read

mode (1 × 50 bp) on an Illumina HiSeq 2500 by the New York University Genome Technology Center. Per-sample FASTQ files were generated using the bcl2fastq2 Conversion software (v2.20) to convert per-cycle BCL base call files outputted by the sequencing instrument into the FASTQ format. Between 30 million and 56 million single-end reads were obtained per sample. The alignment program, STAR (v2.4.5a), was used for mapping reads of 12 human samples to the human reference genome hg19 and the application FastQ Screen (v0.5.2) was utilized to check for contaminants. The software, featureCounts (Subread package v1.4.6-p3), was used to generate the matrix of read counts for annotated genomic features. For the differential gene statistical comparisons between groups of samples contrasted by TIF-IA, UBF, Control, dsDNA, and non-dsDNA conditions, the DESeq2 package (Bioconductor v3.3.0) in the R statistical programming environment was utilized.

### Gene ontology and transfac analysis

Transfac analysis (*Wingender et al., 1996*) was performed using the GATHER gene annotation tool hosted by the Texas Medical Center (*Chang and Nevins, 2006*). Gene ontology analysis was performed using the David Bioinformatics Research 6.8 platform (*Huang et al., 2009a*; *Huang et al., 2009b*).

### siRNA transfections

siRNA transfections were performed as previously described (*Bianco and Mohr, 2017*) using the siRNAs listed in *Supplementary file 4*.

## Acknowledgements

We thank members of the Mohr laboratory and Angus Wilson for helpful discussions and Hannah Burgess, Stephanie Patchett, Elizabeth Vink and Dan Depledge for critically reading the manuscript. This work was supported by National Institutes of Health grants GM056927 and AI073898 to IM. CB was supported in part by Public Health Service Institutional Research Training Award AI07647 and AI007180.

## Additional information

### Funding

| Funder | Grant reference number | Author |
|---|---|---|
| National Institute of General Medical Sciences | GM056927 | Christopher Bianco Ian Mohr |
| National Institute of Allergy and Infectious Diseases | AI073898 | Ian Mohr |
| National Institute of Allergy and Infectious Diseases | AI07647 | Christopher Bianco |
| National Institute of Allergy and Infectious Diseases | AI00718 | Christopher Bianco |

The funders had no role in study design, data collection and interpretation, or the decision to submit the work for publication.

### Author contributions

Christopher Bianco, Conceptualization, Data curation, Formal analysis, Investigation, Visualization, Methodology, Writing - original draft, Writing - review and editing; Ian Mohr, Conceptualization, Formal analysis, Supervision, Funding acquisition, Visualization, Writing - original draft, Project administration, Writing - review and editing

### Author ORCIDs

Christopher Bianco https://orcid.org/0000-0002-5253-4042
Ian Mohr https://orcid.org/0000-0002-4921-1998

**Decision letter and Author response**
Decision letter https://doi.org/10.7554/eLife.49551.sa1
Author response https://doi.org/10.7554/eLife.49551.sa2

## Additional files

### Supplementary files

• Supplementary file 1. List of genes induced by dsDNA in NHDFs treated with non-silencing siRNA, TIF-IA siRNA, or UBF siRNA.

• Supplementary file 2. List of genes regulated by TIF-IA depletion in NHDFs.

• Supplementary file 3. List of primers sequences used for RT-qPCR in this study.

• Supplementary file 4. List of siRNA sequences used in this study.

• Transparent reporting form

### Data availability

All sequencing data generated during this study are available from the sequence read archive (SRA) under the BioProject ID PRJNA528082.

The following dataset was generated:

| Author(s) | Year | Dataset title | Dataset URL | Database and Identifier |
|---|---|---|---|---|
| Bianco C, Mohr I | 2019 | Impact of TIFIA and UBF depletion on genome wide responses to dsDNA | https://www.ncbi.nlm.nih.gov/bioproject/PRJNA528082 | SRA BioProject, PRJNA528082 |

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
