## [Decision Letter]

**Acceptance summary:**

The paper describes unexpected and interesting observations. The authors document the novel findings that reducing ribosome biogenesis in human cytomegalovirus (HCMV)-infected cells engendered unexpectedly an increase in HCMV reproduction. They show a dramatic increase in 45S rRNA abundance, which correlates with increased rRNA transcription factors during HCMV infection. Blocking ribosome biogenesis impairs the cell innate response. They identified a dsDNA binding factor, HMGB2, a chromatin-associated protein, which promotes cytoplasmic double-stranded dsDNA-sensing by cGAS. This constitutes a novel link between ribosomal biogenesis and the innate response. The critical connection is the correlation between the control of HMGB2 production by TIF-IA and the induction of IFNB1 by HCMV, inasmuch as HMGB2-depletion prevents IFNB1 induction by dsDNA. Thus, HCMV infection engenders an increase in TIF-I, and, thus, an increase in HMGB2, which in turn facilitates sensing viral dsDNA, upstream of STING in human cells.

**Decision letter after peer review:**

Thank you for submitting your article "Regulation of HMGB2 integrates ribosome biogenesis and innate immune responses to DNA" for consideration by *eLife*. Your article has been reviewed by two peer reviewers, and the evaluation has been overseen by a Reviewing Editor and Tadatsugu Taniguchi as the Senior Editor. The following individual involved in review of your submission has agreed to reveal their identity: Daphne C Avgousti (Reviewer #2).

The reviewers have discussed the reviews with one another and the Reviewing Editor has drafted this decision to help you prepare a revised submission.

Summary:

The paper describes unexpected and interesting observations. The experiments were expertly done and the conclusions are generally justified. The authors documented a dramatic increase in 45S rRNA abundance, which correlated with increased rRNA transcription factors during HCMV infection. Blocking ribosome biogenesis impaired the cell innate response. They identified a dsDNA binding factor, HMGB2, a chromatin-associated protein, which promotes cytoplasmic double-stranded dsDNA-sensing by cGAS. This constitutes the link between ribosomal biogenesis and the innate response. The critical connection is the correlation between the control of HMGB2 production by TIF-IA and the induction of IFNB1 by HCMV, inasmuch as HMGB2-depletion prevents IFNB1 induction by dsDNA. Thus, HCMV infection engenders an increase in TIF-IA and thus an increase in HMGB2, which in turn facilitates sensing viral dsDNA, upstream of STING in human cells.

Essential revisions:

1) The results regarding HMGB2 as the primary factor linking the ribosome biogenesis, HCMV infection and IFN, are not entirely clear. Overall, the paper seems unfocused, with many (albeit well-done) experiments that do not seem to support the central hypothesis of the paper, that HMGB2 'integrates ribosome biogenesis and innate immunity'. HMGB2 only appears at the end of Figure 6 and in Figure 7, and is not the focus of 75% of the paper, making the title somewhat misleading. It seems that there are two distinct stories wrapped together in this paper: 1 – the unexpected role of ribosome biogenesis in HCMV, which is interesting, convincing and important; and 2 – HMGB2 as a sensor implicated in p53 and IFN signaling. Could other genes, which are disrupted by ribosome biogenesis tampering also be relevant to this link? It is not entirely clear why HMGB2 is the focus, nor whether its biochemical properties contribute to the phenotypes observed. Any further experiments (see point #4 below) should be focused on HMGB2 and why it is the relevant factor.

2) The authors show that knockdown of the transcription factor TIF-IA does not affect protein synthesis in infected cells, while in mock-infected protein synthesis is significantly reduced. Why? How is the virus compensating for the reduction in ribosome biogenesis? The polysome profile shows that while the 80S peak is reduced by TIF-IA depletion the polysome levels are not. This needs to be explained.

3) The majority of the experiments are based on depletion (siRNA treatment) of different transcription factors. Would overexpression of TIF-IA result in a more effective innate immune response and suppression of virus replication?

4) Importantly, would overexpression of HMGB2 alleviate the need for TIF-IA activation in response to infection?

---

## [Author Response]

Essential revisions:1) The results regarding HMGB2 as the primary factor linking the ribosome biogenesis, HCMV infection and IFN, are not entirely clear. Overall, the paper seems unfocused, with many (albeit well-done) experiments that do not seem to support the central hypothesis of the paper, that HMGB2 'integrates ribosome biogenesis and innate immunity'. HMGB2 only appears at the end of Figure 6 and in Figure 7, and is not the focus of 75% of the paper, making the title somewhat misleading.

We have revised the title of the manuscript as requested.

It seems that there are two distinct stories wrapped together in this paper: 1 – the unexpected role of ribosome biogenesis in HCMV, which is interesting, convincing and important; and 2 – HMGB2 as a sensor implicated in p53 and IFN signaling.

These “distinct” stories are correlated in Figure 7. This figure demonstrates that i) IFNB1 production in response to HCMV, which we show (Figures 3 and 4) is augmented by ribosome biogenesis, is dependent upon HMGB2 (Figure 7C); and ii) TIF-IA depletion reduces cytoplasmic HMGB2 levels (Figure 7B). We now clarify this in the text (subsection “Interferon induction in response to HCMV requires HMGB2”; Figure 7 legend).

Could other genes, which are disrupted by ribosome biogenesis tampering also be relevant to this link?

We have clarified the Discussion (second paragraph) to acknowledge that other genes repressed upon TIF-IA depletion may similarly influence cell intrinsic innate responses. In addition, we have modified the Abstract to indicate that HMGB2 is one of many genes dysregulated upon interfering with ribosome biogenesis.

It is not entirely clear why HMGB2 is the focus, nor whether its biochemical properties contribute to the phenotypes observed.

Our focus on HMGB2 as a gene repressed upon TIF-IA depletion was based on i) published work from Taniguchi’s lab establishing that HMGB2 plays a critical role in cytoplasmic dsDNA signaling and type I interferon production; ii) the accumulation of p53 in response to TIF-IA-depletion, and repression of HMGB2 gene expression by p53. We have clarified this in the subsection “TIF-IA depletion regulates expression of NFY and p53-responsive DREAM complex target genes including the DNA sensor HMGB2”.

Any further experiments (see point #4 below) should be focused on HMGB2 and why it is the relevant factor.2) The authors show that knockdown of the transcription factor TIF-IA does not affect protein synthesis in infected cells, while in mock-infected protein synthesis is significantly reduced. Why? How is the virus compensating for the reduction in ribosome biogenesis? The polysome profile shows that while the 80S peak is reduced by TIF-IA depletion the polysome levels are not. This needs to be explained.

While these are excellent points, we are unable at present to rigorously and unambiguously explain the underlying molecular mechanism(s) at work (which are beyond the scope of the present study). Nevertheless, we now point this out directly in the text (subsection “Ribosome abundance and protein synthesis can be uncoupled in HCMV infected cells”) and offer possible explanations in the Discussion (last paragraph).

3) The majority of the experiments are based on depletion (siRNA treatment) of different transcription factors. Would overexpression of TIF-IA result in a more effective innate immune response and suppression of virus replication?

Answered below together with point #4.

4) Importantly, would overexpression of HMGB2 alleviate the need for TIF-IA activation in response to infection?

NHDFs were stably transduced with a lentivirus expressing either GFP, TIF-IA, or HMGB2. Transduced cells were enriched by Blasticidin selection Ectopic expression was validated by immunoblotting. We did *not* reproducibly detect reduced expression of HCMV representative immediate early (IE1/IE2) or late (pp28) proteins in NHDFs stably transduced with a TIF-IA expression vector compared to NHDFs expressing GFP, nor did we detect a significant reduction in virus growth in TIF-IA overexpressing cells compared to GFP-expressing NHDFs. In addition, TIF-IA overexpression did not detectably alter p53 or HMGB2 protein levels, which is consistent with our inability to detect a difference in HCMV growth under these conditions. In hindsight, this is not particularly surprising given the greater extent to which HCMV infection *already* stimulates accumulation of RNA pol I transcription factors including TIF-IA. Thus, further overexpression of TIF-IA did not detectably result in a more effective innate immune response capable of suppressing virus reproduction).

Overexpression of HMGB2 did not detectably reverse the reduction in IFNB1 mRNA levels in response to TIF-IA depletion in HCMV infected cells. Importantly, while HMGB2 overexpression was readily observed both by immunoblotting and by indirect immunofluorescence, HMGB2 overexpression only increased *nuclear* HMGB2, but did not detectably raise cytoplasmic HMGB2 levels. It is this cytoplasmic HMGB2 subpopulation that has been implicated in DNA sensing.

We agree had these experiments produced interpretable results, they would have been worth including in the study. Unfortunately, both experiments involved ectopic overexpression of target proteins in primary NHDFs and yielded negative results consistent with a “failure to find”. Given these findings, a meaningful interpretation of the results is not forthcoming. Numerous factors could potentially account for this, not the least of which are the inherent difficulties in achieving proper dose expression levels and the unknown consequences of creating a stably-transduced population of primary cells expressing the target gene needed for experimental analysis. For these reasons, we believe the *positive r*esults we present in our manuscript where we transiently create hypomorphic alleles using multiple, independent siRNAs represents a more tractable, powerful approach to dissecting gene function.